# Live Birth Rate in Women with Recurrent Pregnancy Loss after In Vitro Fertilization with Concomitant Intravenous Immunoglobulin and Prednisone

**DOI:** 10.3390/jcm11071894

**Published:** 2022-03-29

**Authors:** Pia Egerup, Henriette Svarre Nielsen, Anders Nyboe Andersen, Ole Bjarne Christiansen

**Affiliations:** 1The Recurrent Pregnancy Loss Unit, Capital Region, Copenhagen University Hospitals, Hvidovre Hospital and Rigshospitalet, Blegdamsvej 9, 2100 Copenhagen, Denmark; piaegerup@hotmail.com (P.E.); henriette.svarre.nielsen@regionh.dk (H.S.N.); 2Department of Obstetrics and Gynecology, Copenhagen University Hospital Hvidovre, Kettegård Alle 30, 2650 Hvidovre, Denmark; 3Department of Clinical Medicine, University of Copenhagen, 2200 Copenhagen, Denmark; 4The Fertility Department, Rigshospitalet, Copenhagen University Hospital, Blegdamsvej 9, 2100 Copenhagen, Denmark; anders.nyboe.andersen@regionh.dk; 5Department of Obstetrics and Gynecology, Aalborg University Hospital, Reberbansgade 15, 9000 Aalborg, Denmark

**Keywords:** recurrent pregnancy loss, IVF, intravenous immunoglobulin, prednisone

## Abstract

Pregnancy loss after in vitro fertilization (IVF) is at least as common as after spontaneous conception. Recurrent pregnancy loss (RPL) may often have an immunological background, and it is therefore relevant to test immune-based interventions in these patients. The objective was to investigate the effect of immunotherapy with intravenous immunoglobulin (IvIg) and prednisone (PRS) as concomitant therapy to IVF in women with RPL after earlier IVF treatments. In a cohort study conducted at The Danish RPL Clinic, 41 women with three or more consecutive pregnancy losses after IVF underwent at least one further IVF cycle with concomitant immunotherapy from 2012 to 2017. The immunotherapy with IvIg and PRS was given before embryo transfer and repeatedly in the first trimester when pregnancy was achieved. Fourteen women (34.2%) achieved a live birth after the first embryo transfer with immunotherapy, and a total of 32/41 (78%) achieved a live birth after up to 4 embryo transfers. Baseline characteristics and the presence of autoantibodies were not significantly different among women achieving live birth or not. The observed 34% birth rate in women with RPL after IVF receiving immunotherapy appears higher than the expected 16–19% birth rate without immunotherapy and is similar to findings in a previous cohort from our clinic. Concomitant immunotherapy as described may be a promising intervention for women with RPL after IVF; however, the effect must be tested in a randomized controlled trial.

## 1. Introduction

In vitro fertilization (IVF) and intra-cytoplasmatic sperm injection (ICSI) are the key treatments in modern management of infertility. Pregnancy loss after IVF is at least as common as after spontaneous pregnancies [1,2], and based on National Register data from the United States additionally 15% of all intrauterine pregnancies after assisted reproductive technology (ART) are lost during the rest of the first trimester [3]. One pregnancy loss after ART may slightly improve the prognosis for later live birth [4]. However, in the case of recurrent pregnancy loss (RPL) after ART, little is known concerning the subsequent chance for live birth. Whereas register data are based on miscarriages occurring after establishment of a clinical pregnancy, pregnancy losses also comprise biochemical pregnancies losses, which are not included in national registers [5].

Different autoantibodies can be detected with increased frequency in women with RPL, and carriage of certain human leukocyte antigen (HLA)-DR and -DQ alleles is associated with RPL [6,7]. Associations between a disease and particular HLA-DR/DQ alleles are in general thought to indicate an immunological background. It is therefore relevant to test immune-based interventions in RPL patients.

Intravenous immunoglobulin (IvIg) has several effects, such as inhibition of complement binding, modification of cytokine production, attenuation of natural killer cells, suppression and neutralization of autoantibodies, and expansion of regulatory T lymphocytes. IvIg exhibits a documented effect in many disorders caused by immunological abnormalities and has been tested in randomized trials with RPL patients [8]. Two recent systematic reviews suggest that IvIg may be effective in women with secondary RPL or if the treatment is initiated before pregnancy [8,9]. Immunotherapy with peri-implantation glucocorticoid administration (e.g., prednisone (PRS)) has been associated with an insignificant trend towards an increase in pregnancy rates in a subgroup of IVF patients but not in ICSI patients [10]. IvIg has proved to be well-tolerated in pregnant women without any increased rate of serious adverse events (e.g., death, hospitalisation, caesarean section, fetal growth retardation) for the treated women or the children in randomized controlled trials of RPL women [8]. However, a minor increase in mild adverse events has been noted, e.g., rash, itching, fever, headaches [8]. PRS has been used in pregnant RPL women in doses up to 20 g daily from positive pregnancy test until week 12 without evidence of excess risk for the women or the fetus compared with untreated patients [11], but in the present study we chose a PRS dose of 10 mg primarily for safety reasons because we started treatment already four weeks before a positive pregnancy test.

At the RPL Clinic at the Fertility Clinic, Rigshospitalet, Copenhagen, Denmark, women with RPL after IVF/ICSI treatment have, since 2003, routinely been offered treatment with IvIg and PRS in the IVF/ICSI cycle preceding pregnancy and in early pregnancy if the woman conceived. In 2014, we reported that 36.5% had a live birth after this treatment regimen [12]. The study included, however, only 52 patients, and was thus of too limited a sample size to evaluate the efficacy of the immunotherapy. We therefore conducted the present follow-up study as a confirmatory study to examine if our earlier reported favorable results with IvIg and PRS were a robust finding also in a later study period. We therefore report live birth rates in 41 women with RPL who all had three or more IVF/ICSI pregnancy losses and therefore were treated with IvIg and PRS from 2012–2017 as an adjunct to subsequent IVF/ICSI cycles.

## 2. Material and Methods

### 2.1. Patients

All patients referred to the RPL Clinic at the Fertility Clinic, Rigshospitalet, University Hospital of Copenhagen, Denmark, were recorded in a detailed database for research purposes. The database contained information on previous pregnancies, previous fertility treatments, blood samples and fertility treatments in the clinic including concomitant treatments. From this, database we were able to identify and include patients who met the following inclusion criteria:(1)Referred between 12 January 2012 to 1 May 2017;(2)Before referral: three or more consecutive pregnancy losses (includes both miscarriages and biochemical pregnancy losses) after transfer of fresh or frozen embryos obtained by IVF/ICSI treatment with the same partner;(3)After referral: at least one IVF or ICSI cycle that resulted in embryo transfer with the same partner in which IvIg and PRS were administered.

All pregnancies prior to and after inclusion were documented by either positive urine- or serum hCG, transvaginal ultrasound examination, or histology after curettage. Confirmed ectopic pregnancies were not included.

All couples underwent a screening program for RPL that included screening for parental chromosome aberrations, maternal HLA class II typing, autoantibodies (IgM and IgG anticardiolipin antibodies, lupus anticoagulant, anti-double-stranded-DNA antibodies and anti-thyroidperoxidase antibodies), anti-Müllerian hormone (AMH) and anatomic abnormalities detected by hysteroscopy, hysterosalpingography or hydrosonography. The investigations for uterine abnormalities only aimed to evaluate the anatomy of the uterus and did not include any endometrial biopsies. Patients with parental chromosome aberrations and significant anatomic abnormalities were excluded from the study. Women with autoantibodies were not excluded from the study.

### 2.2. Immunomodulatory Treatment

All patients included in the study received one intravenous infusion with IvIg preconceptionally. Immunoglobulin Privigen^®^ 100 mg/mL (CSL Behring, Bern, Switzerland), 25 g in total or from 2014 and onwards 0.4 g per kg, was administered from 5 days before to 1 day after embryo transfer (ET). Additionally, 10 mg oral PRS was given daily from the first day of down-regulating in IVF/ICSI cycles stimulated with the long agonist protocol and from the first day of FSH administration in IVF/ICSI cycles using the antagonist protocol. In frozen ET (FET) cycles, prednisone was started on the first day of the cycle. None of the women in our study received any other kind of immunotherapy than immunoglobulin and PRS.

In patients with a positive serum hCG after ET and a second measurement showing ≥30% increase per 24 h, repeated IvIg infusions of 25 g or 0.4 g per kg were initiated. Infusions were given weekly from the 5–9 gestational week and thereafter at 2-weeks intervals to gestational week 14 (total 7 infusions), if repeated transvaginal ultrasound confirmed a viable intrauterine pregnancy. PRS was continued until gestational week 7 and then stopped after stepwise reduction over three days (half dose (5 mg daily for 3–4 days) and then stopped), after advice from endocrinologists. If the pregnancy was ongoing after gestational week 15, monitoring in our clinic was stopped and the patients were referred to their local hospital for further follow-up. The follow-up until 15 weeks in our clinic is due to a low risk of pregnancy loss after this time (less than 1%), but we are aware of the lack of evidence regarding the timing for the safe withdrawal of close monitoring. Data regarding obstetrical outcomes were collected from questionnaires that were completed and returned by the patients after the birth.

If the patient did not become pregnant in the first cycle with preconceptional IvIg, no further IvIg infusions were given if the next ET was performed within three months. If the patient did not become pregnant within three months, a new preconceptional IvIg infusion was given. In addition, PRS was administered in subsequent ET cycles.

### 2.3. Statistics

The χ^2^-test were used for comparing binary variables: presence of autoantibodies, infertility causes and number of patients with low AMH. The Wilcoxon–Mann–Whitney U-test was used for comparing continuous variables: age at referral, body mass index, number of previous pregnancy losses, number of ART cycles with negative serum hCG before referral, total number of unsuccessful ETs before referral and number of ART cycles with immunomodulation after referral.

All analyses were performed using R version 3.4.3 (www.R-project.org, The R Foundation).

### 2.4. Ethics

The reported treatment with IvIg and PRS has been the standard treatment for women with RPL after IVF/ICSI since 2003 at the RPL Clinic at the Fertility Clinic, Rigshospitalet. The treatment is offered to women with ≥3 confirmed consecutive pregnancy losses (including both biochemical pregnancy losses and miscarriages) after IVF/ICSI treatment. The women were not selected due to immunological biomarkers including autoantibodies. The treatment with IvIg and PRS is free of charge for the patients, as the treatments is administrated at the public RPL Clinic. Thus, no approval from the local ethics committee was sought. The Recurrent Pregnancy Loss Database provided data for the study, which was approved by the Danish Data Protection Agency for Research (journal number: 2012-58-0004/RH-2017-315 I-Suite 05939). Access to the patients’ personal electronic files was approved by The Danish Patient Safety Authority (journal number: 31-1521-424).

## 3. Results

A total of 41 patients with RPL after IVF/ICSI received treatment with IvIg and PRS in the study period. Figure 1 provides a flowchart illustrating the outcome after the first, second, third and fourth ET with IvIg and PRS.

Fourteen (34.2%) of the included 41 patients had a live birth after the first treatment with IvIg and PRS. A total of 32 patients achieved a live birth after a maximum of four ETs, corresponding to a crude observed live birth rate of 78.0%. The overall live birth rate per embryo transfer among all patients in the study group was 42.7% (32 live births after 75 embryo transfers; see Figure 1). Several women were treated at various private clinics. Information on number of embryos transferred were only available to us from patients treated at public clinics, and this accounted for 22 ET’s; 10 were single-embryo transfer (SET) and 12 were double-embryo-transfer (DET). A day-2 embryo was transferred in 8/22 ET’s and a blastocyst in 14/22 ET’s. There were no ETs after pre-implantation genetic testing for aneuploidies (PGT-A) because this testing was not available in Denmark before 2017 and none had undergone pre-implantation genetic testing for structural rearrangements (PGT-SR) because all couples had normal parental chromosomes.

The pregnancy loss rates after 1st, 2nd, 3rd or 4th ETs were 39.1% (9/23), 26.7% (4/15), 0% (0/6) and 0% (0/1), respectively (Figure 1). A total of 13 of the 45 achieved pregnancies that ended in a pregnancy loss, corresponding to a crude observed pregnancy loss rate of 29%. The 13 pregnancy losses included 6 biochemical losses, 6 miscarriages and 1 ectopic pregnancy.

Table 1 shows the characteristics of the patients achieving live birth or no live birth, respectively, after up to four ART cycles with immunotherapy. Overall, only 5/41 patients were positive for the investigated autoantibodies (three women had anti-thyroidperoxidase antibodies and two women had anti-double-stranded-DNA antibodies). There were no significant differences according to age at referral, body mass index (BMI), presence of autoantibodies and number of patients with low AMH. The patients with live birth had a median of 3.5 previous pregnancy losses before referral, and the patients with no live birth had a median of 4.0 losses (*p* = 0.37). The median total number of embryo transfers without live births (no pregnancy achieved or pregnancy loss) before immunotherapy was 4.0 in both groups. As seen in Table 1, the distribution of the different infertility diagnoses was not significantly different in the two groups. The patients underwent a median of two ART cycles treated with IvIg and PRS.

In Table 2, we have combined the data from our previous study [12] with the present dataset, as both studies used the same inclusion criteria and same type of immunotherapy. Table 2 shows that, overall, 64 live births were obtained in 93 patients, corresponding to a crude observed live birth rate of 68.8%.

## 4. Discussion

In this study including 41 women with RPL after IVF/ICSI, we found a live birth rate of 34.2% after the first ET that was combined with IvIg and PRS as adjunct treatment. The crude observed, cumulative live birth rate after up to four treatments that resulted in ETs was 78.0%. Comparing women with live birth and women without live birth, there were no differences regarding age, BMI, low AMH, presence of autoantibodies, infertility cause, number of previous pregnancy losses, number of ART cycles with negative serum hCG or number of unsuccessful ETs before referral.

The observed pregnancy loss rate was 29%. The pregnancy loss rate after the first ET was 39.1%, which then decreased with increasing number of ETs with immunotherapy. This decrease may be due to more immunotherapies with more treatments, increasing the immunomodulating effect. However, the results should be interpreted with caution in relation to the small number of women in the third and fourth ETs. The 29% crude observed pregnancy loss rate observed in this study is similar to the general loss rate after IVF observed in large good clinical practice (GCP) monitored multicenter trials [1,2]. We would, however, expect the pregnancy loss rate to be higher in our population because the included women already had three of more losses after IVF. The fact that the pregnancy loss rate corresponds to the general loss rate after IVF therefore may indicate that immunotherapy with IvIg and PRS decreases the pregnancy loss rate and thereby improves live birth rates.

We are aware that the number and stage of embryos transferred in the cycles are important for the success rate after IVF/ICSI. We were granted permission from the Danish Patient Safety Authority to achieve information on these laboratory data from our own fertility clinic but no other clinics, and we were thus only able to get information on 22 ETs performed. We think the distribution of day-2 and blastocyst transfers as well as SETs and DETs are typical for the time before 2017. Due to the limited number of ETs with information about embryos we did not make any analysis of outcome in our cohort adjusted for these parameters.

We only included patients with ≥3 consecutive pregnancy losses, which of cause restricts the number of patients possible to include in the study. However, this is a group of patients with a poorer spontaneous prognosis than those with fewer or non-consecutive pregnancy losses and they are in our view more informative as regards the effect of the treatment.

Previous studies of immunomodulatory treatment of women with recurrent IVF/ICSI failure [13,14,15,16,17] have failed to define what is meant with IVF/ICSI failure: a negative hCG after embryo transfer only (real implantation failure) or an embryo transfer followed by a negative hCG test or a biochemical pregnancy loss or miscarriage. In our cohorts we have, for the first time, selected patients due to the strict criteria of a minimum of three consecutive pregnancy losses after IVF/ICSI because we believe that the pathophysiological background for implantation failure and early pregnancy losses in many cases is different, with immunological factors playing a more prominent role in the latter cases. We counted previous biochemical pregnancies as pregnancy losses in the patients´ reproductive histories and thus in the selection for the study because a previous cohort study in RPL patients from our clinic has reported that biochemical pregnancy losses exhibit a similar negative prognostic impact on subsequent pregnancy outcome as clinical pregnancy losses [18]. In addition, the RPL Guidelines of the European Society of Human Reproduction and Embryology (ESHRE) recommends that biochemical (non-visualized) losses should be included in the criteria for getting the diagnosis RPL [19].

The previous study from our clinic with comparable patients from 2003–2012 showed a live birth rate of 36.5% at first IVF/ICSI treatment with IvIg and PRS and an observed cumulative live birth rate of 61.5% [12]. Overall, the very similar results in these two cohort studies indicate that IvIg and PRS after the first treatment is associated with a live birth rate per transfer cycle of approximately 35.5% in women with RPL (Table 2). Secondly, a combined crude observed live birth rate of approximately 68.8% can be expected after several attempts (Table 2).

The expected live birth rate after ≥3 failed IVF/ICSI attempts (started cycles) is from the literature assumed to be approximately 16–19% [20,21], but the estimate is uncertain. With the reservation that patients in the present study only included those that had embryo transfer (ET), the results still seem favourable. It therefore seems that IvIg and PRS may increase the live birth rate substantial and may thus be a promising treatment for women with RPL after IVF/ICSI.

In accordance with our study, a recent review [22], which summarizes evidence from mainly non-randomised studies [12,13,14,15,16,17,23,24,25,26] found a live birth rate of approximately 40% after IvIg treatment in patients with RPL/recurrent implantation failure after IVF/ICSI. It is, however, important to note that RPL and recurrent implantation failure are heterogeneous conditions, and it is therefore not possible to make direct comparisons with our study.

Our rationale for testing immunomodulating therapy in women with RPL after IVF treatment is based on several indications that immune disturbances are involved in both unsuccessful IVF [27,28,29,30] and unexplained RPL [7,31,32,33,34,35]. Our previous study [12] supports this argument, and we here present further support.

The present study provides no information regarding how IvIg (and PRS) may work in the prevention of pregnancy loss. Based on other studies, we hypothesize that IvIg causes an increased live birth rate in women with RPL (with or without IVF) by upregulating regulatory T cells (Tregs) that induce tolerance [36], and/or down-regulate the number of natural killer (NK) cells [14] or the proportion of CD3+/CD56 NK cells [37] and thereby modify the immune response to embryonal or trophoblast antigens. The IvIg doses of initially 25 g and, later in the study period, 0.4 g per kg and the infusion intervals of first one week and after gestational week 9 every second week were chosen on a purely empirical basis. We adhered to this protocol because it was associated with an apparently good reproductive outcome and very few side effects. We tested the combined effect of IvIg and PRS because, in several autoimmune diseases, the combined effect of these drugs exceeds that of each drug as monotherapy. A good example is severe autoimmune thrombocytopenia where the combination treatment of IvIg and PRS is the golden standard because it has proved most effective [38]. However, more clinical trials and basic research are needed to find the most cost-effective treatment protocol and explore the exact mechanisms of IvIg and PRS in the RPL population.

We have not excluded women with autoantibodies from the study as we consider the presence of autoantibodies as only one of many markers of a complete or partial immunological background for failed IVF treatments. There are probably many immunological disturbances of importance for IVF failure, which are not associated with positive testing for autoantibodies, such as dominance of proinflammatory cytokines or increased NK cell cytotoxicity activity in the uterus. Ledee et al. found, for example, that 82% of recurrent implantation failure (RIF) patients had immunological dysregulation in the endometrium assessed by analysis of IL-15/Fn-14 mRNA and IL-8/TWEK mRNA ratios [39]. Another example of immune disturbances, which are not related to autoantibodies but may be associated with RPL, has been reported by Zhu et al. [40], who found that TGF-beta positive NK and T lymphocytes characterize RPL patients compared to controls. As the investigation of these and other immunological biomarkers is still reserved for very few specialized laboratories, and/or their importance for IVF failure and RPL for the time being has not been sufficiently validated, we have chosen in this study not to select patients based on immunological biomarkers. This decision not to select patients to immunotherapy based on presence of specific immunological biomarkers is in accordance with the recommendation in the ESHRE Guideline for RPL that “no immunological biomarker, except for high-titer antiphospholipid antibodies, can be used for selecting couples with RPL for specific immunological treatments” [19].

The present study is a cohort study without an untreated control group, and the findings should therefore be interpreted with caution. Randomized controlled trials are thus needed to document the effect of IvIg and PRS in women with RPL after IVF/ICSI. Based on the positive results in this follow-up study, we have, in 2021, initiated a randomized controlled trial, testing IvIg and PRS against placebo (human albumin and oral placebo).

## 5. Conclusions

The 34% live birth rate in women with RPL after IVF/ICSI receiving IvIg and PRS appears higher than the 16–19% live birth rate reported in other cohorts of women with recurrent IVF/ICSI failure, and it seems that the pregnancy loss rate is comparable with non-RPL patients after IVF. Immunotherapy with IvIg and PRS may thus be a promising treatment for women with RPL after IVF/ICSI, but the efficacy should be evaluated in a randomized controlled trial.

## Figures and Tables

**Figure 1 jcm-11-01894-f001:**
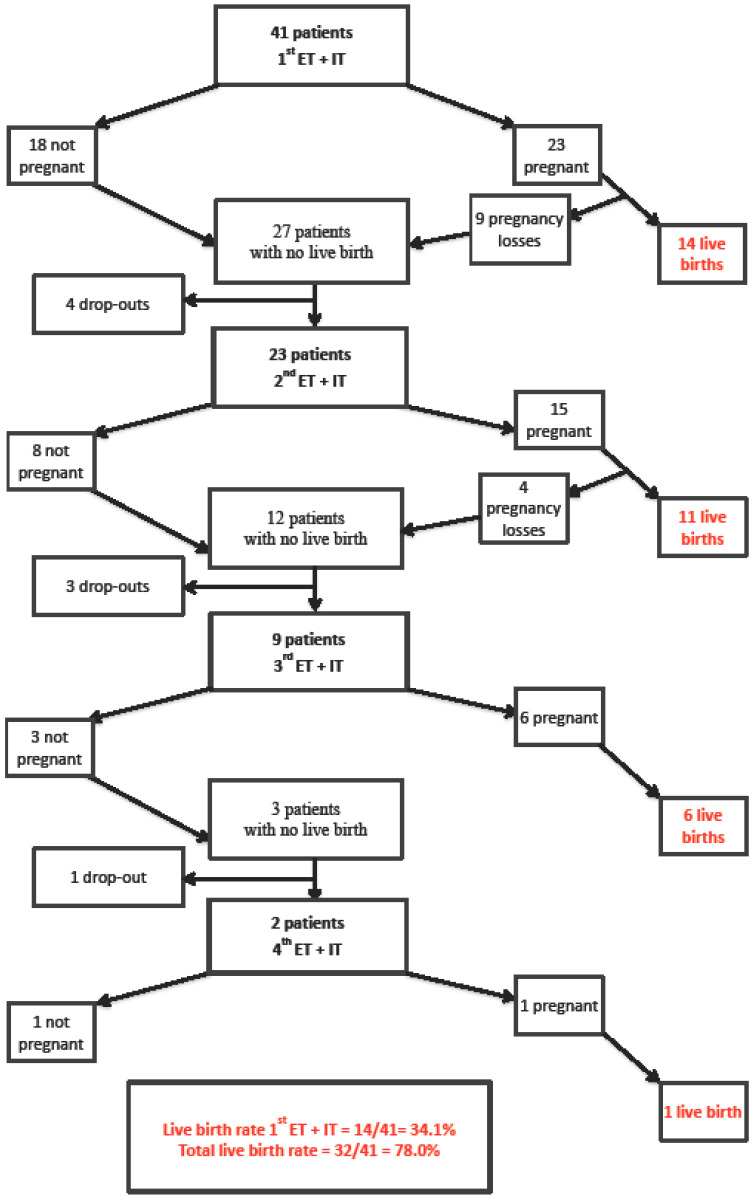
Outcome after first, second, third and fourth embryo transfer (ET) with immunotherapy (IT) with prednisone and intravenous immunoglobulin.

**Table 1 jcm-11-01894-t001:** Characteristics of the women with live birth and no live birth, respectively.

Data Collected	No Live Birth (*n* = 9)	Live Birth (*n* = 32)
Age at referral (median (IQR))	34.00 [29.00, 36.00]	35.00 [33.00, 38.00]
Body Mass Index (median (IQR))	23.50 [21.75, 26.25]	22.00 [20.75, 26.00]
Presence of autoantibodies ^1^, (*n* (%))	1 (11.1)	4 (12.5)
AMH < 5 pmol/L (*n* (%))	0 (0.0)	2 (6.2)
Infertility cause (*n* (%))		
Tuba factor	2 (22.2)	2 (6.2)
Uterine factor	0 (0.0)	1 (3.1)
Male factor	2 (22.2)	12 (37.5)
Endometriosis	0 (0.0)	2 (6.2)
Anovulation/PCOS	1 (11.1)	2 (6.2)
Egg factor	1 (11.1)	3 (9.4)
Unexplained	1 (11.1)	3 (9.4)
Mixed	2 (22.2)	7 (21.9)
No. of pregnancy losses before referral ^2^ (median (IQ))	4.00 [3.00, 4.00]	3.50 [3.00, 4.00]
ART cycles with negative serum hCG before referral (median (IQR))	3.00 [1.00, 5.00]	1.00 [0.00, 3.25]
Total no. of unsuccessful ET´s before referral ^3^ (median (IQR))	4.00 [3.00, 9.00]	4.00 [4.00, 6.25]
ART cycle with immunomodulation after referral (median (IQR))	2.00 [1.00, 2.00]	2.00 [1.00, 2.00]

^1^ Thyroid peroxidase antibody, IgG anticardiolipin antibody, lupus anticoagulant, antinuclear antibody, anti-ds-DNA antibody. ^2^ Inclusive spontaneous conception and ART. ^3^ Exclusive confirmed ectopic pregnancies and induced abortions.

**Table 2 jcm-11-01894-t002:** Results from the two studies conducted at the clinic regarding immunotherapy (IvIg and PRS) to women with RPL after IVF/ICSI and the combined live birth rate.

	Nyborg et al. 2014	Egerup et al. 2022	Combined
Number of women	52	41	93
Live birth rate after the first embryo transfer	36.5% (19/52)	34.1% (14/41)	35.5% (33/93)
Crude observed live birth rate	61.5% (32/52)	78.0% (32/41)	68.8% (64/93)

## Data Availability

It is not possible to share data due to personal data.

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
