# Peer review of "Live Birth Rate in Women with Recurrent Pregnancy Loss after In Vitro Fertilization with Concomitant Intravenous Immunoglobulin and Prednisone"

_jcm, 2022, doi:10.3390/jcm11071894_

Round 1
Reviewer 1 Report
I congratulate the authors for their determination to conduct a study in a setting where there is a significant lack of scientific evidence (autoimmunity and infertility). As they themselves say during the manuscript, the study group is small, and there is no control group (either with another drug or with placebo). It would be very interesting to be able to compare them.
Author Response
We thank the reviewer for the positive comments. We hope to able to publish a placebo-controlled trial on the subject within a two years
Reviewer 2 Report
The authors present a retrospective cohort study to evaluate live birth rate (LBR) after intravenous immunoglobuline and prednisone treatment in couples with recurrent pregnancy loss after IVF.
It is a retrospective study with no control group and all the limitations of this study design. I would like to congratulate the authors for initiating a prospective study with a control group to further investigate this important question.
Furthermore two different immunomodulatory therapies were combined in a group of patients without any testing or diagnosis of an immunological disorder.
I recommend to explain in more detail why two different therapies were mixed, why biochemical pregnancies were included, and why there was no testing on other causes for RPL.
The introduction mentions administration of 20 g prednisone. Surely 20 mg are meant. Please elaborate on the dosage, as in the current study 10 mg are used.
Author Response
We combined two different immunomodulatory therapies: IvIg and prednisolone since in many autoimmune/inflammatory diseases (e.g. autoimmune thrombocytopenic purpura) these therapies are very often combined and seem to potentiate each other. We have now made a reference to this (page 14-15).
We included/accepted biochemical pregnancies both in the criteria for getting the diagnosis of recurrent pregnancy loss and also as an outcome in the study.. This is based both on data from our own clinic that biochemical pregnancies exhibit the same negative prognostic impact in recurrent pregnancy loss patients as clinical miscarriages (Kolte et al. 2014) and also because the ESHRE Recurrent Pregnancy Guideline (2018) recommends that biochemical pregnancies should count in the criteria for the diagnosis of recurrernt pregnancy loss. This is now mentioned in page 13.
We did not select our patients to immunmodulatory treatment due to finding some immunological abnormality in peripheral blood. We are convinced that in most cases of recurrent pregnancy loss, the immunological reactions responsible for embryonal death is taking place in the uterus or regional lymph nodes and peripheral blood biomarkers are not or only weakly reflecting what happens in the uterus. In the ESHRE Guidelines for Recurrent Pregnancy Loss is stated that "no immunological biomarker except for high-titer antiphospholipid antibodies, can be used for selecting couples with recurrent pregnancy loss for specific immunological treatments". We agree with this statement and therefore made to selection due to immune biomarkers except for antiphospholipid antibodies. None of our patients were positive for anti-phospholipid antibodies.
In the introduction (page 4) we clarify that we chose a lower dose of prednisolone (10 mg daily) than in the Tang et al. (2013) study primarily of saftely reasons since we were treating the patients for a longer timer (from 4 week before a positive pregnancy test) whereas in the Tang study treatment was only starte in early pregnancy
Reviewer 3 Report
In this article the authors analysed the possible effects on live birth rate of intravenous immunoglobuline and prednisone in patients with recurrent pregnancy loss. The author concluded that women who received immunotherapy had a live birth rate of 34%, higher than the 16-19% expected in patients with recurrent pregnancy loss without immunotherapy. These findings could be very interesting, but we have to remark on some points:
- The main weakness of this study, as also partly stated by the authors themselves, is the small number of patients recruited, only 41, and the purpose of the author to investigate on the live birth rate.
- Another possible flaw of the study is the lack of information about the embryos that were transferred. The authors doesn't provide neither information about the number of embryos transferred per cycle nor details about the stage of the embryos. It is also not specified if PGT was performed.
- It should be added among the characteristic of the patients if a hysteroscopy or an endometrial biopsy was performed, due to the fact that this cohort of patients had recurrent pregnancy loss
Author Response
Thank you for good comments.
We agree that the number of included patients are limited but since we adhered to the traditional diagnostic criteria of recurrent pregnancy loss: 3 or more consecutive pregnancy losses. This is the number that it is possible to collect in the largest Scandinavian fertility clinic receiving patients from most of Denmark in a 5 years period. Only our previous study (Nyborg et al. 2014) has been able to collect a publish a larger group of these quite rare patients.
Since the patients were referred from many fertility clinics in Denmark among them several private clinics, information about stages and numbers of embryos transferred could often not be collected from these clinics since it would require multiple permissions from the Danish Patient Safety Authority. We got permission to get the data from the public clinics and obtained data on 22 embryo transfers (page 8-9). We believe that the distribution of transfers of day-2 embryos and blastocysts and of single and double embryo transfers are representative for the time period in our country. In no case was PGT-A or PGT-SR performed and whereas all patients had normal uterine cavities by hysteroscopy. hysterosalpingography or hydrosonography, none had emdometrial biopsies taken (page 5-6)